# Cost-effectiveness of conditional cash transfers to retain women in the continuum of care during pregnancy, birth and the postnatal period: protocol for an economic evaluation of the Afya trial in Kenya

Neha Batura ,[1] Jolene Skordis ,[1] Tom Palmer,[1] Aloyce Odiambo,[2] Andrew Copas,[1] Fedra Vanhuyse,[3] Sarah Dickin,[3] Alie Eleveld,[2] Alex Mwaki,[2] Caroline Ochieng,[3] Hassan Haghparast-Bidgoli [1]

[1]Institute for Global Health, University College London, London, UK
[2]Safe Water and AIDS Project, Kisumu, Kenya
[3]Stockholm Environment Institute, Stockholm, Sweden

**Correspondence to**
Dr Neha Batura;
n.batura@ucl.ac.uk

## ABSTRACT

**Introduction** A wealth of evidence from a range of country settings indicates that antenatal care, facility delivery and postnatal care can reduce maternal and child mortality and morbidity in high-burden settings. However, the utilisation of these services by pregnant women, particularly in low/middle-income country settings, is well below that recommended by the WHO. The Afya trial aims to assess the impact, cost-effectiveness and scalability of conditional cash transfers to promote increased utilisation of these services in rural Kenya and thus retain women in the continuum of care during pregnancy, birth and the postnatal period. This protocol describes the planned economic evaluation of the Afya trial.

**Methods and analysis** The economic evaluation will be conducted from the provider perspective as a within-trial analysis to evaluate the incremental costs and health outcomes of the cash transfer programme compared with the status quo. Incremental cost-effectiveness ratios will be presented along with a cost-consequence analysis where the incremental costs and all statistically significant outcomes will be listed separately. Sensitivity analyses will be undertaken to explore uncertainty and to ensure that results are robust. A fiscal space assessment will explore the affordability of the intervention. In addition, an analysis of equity impact of the intervention will be conducted.

**Ethics and dissemination** The study has received ethics approval from the Maseno University Ethics Review Committee, REF MSU/DRPI/MUERC/00294/16. The results of the economic evaluation will be disseminated in a peer-reviewed journal and presented at a relevant international conference.

**Trial registration number** NCT03021070

## Strengths and limitations of this study

► The protocol describes planned data collection and analyses for economic analyses, which can aid evaluations of public health interventions in similar contexts.
► The protocol demonstrates an application of new global guidelines for economic evaluations.
► The study design will contribute to our understanding of methods to evaluate cost, cost-effectiveness and cost consequences of conditional cash transfer programmes.
► The study design contributes to our understanding of methods to evaluate fiscal space for investments in maternal and child health in resource-constrained settings.
► The data collection methods proposed for this study may need adaptation before use in other settings.

to delivery, the immediate postnatal period and early childhood.[1] WHO guidance on routine focused antenatal care (ANC) for pregnant women recommends eight points of contact with health services. However, in sub-Saharan Africa (SSA), only 44% of women attend four recommended visits that constitute focused ANC.[2] This could compromise the effectiveness of care, decreasing the likelihood of positive pregnancy outcomes.[3] Further, ~50% of postnatal maternal deaths occur during the first week after delivery and one in four child deaths occur in the first month of life, meaning that postnatal care (PNC) is equally crucial.[4] In SSA, PNC programmes are among the weakest of all reproductive and child health programmes.[5]

## INTRODUCTION
### Background

A key strategy to improve maternal and child health is to ensure continuity of care for mothers and their babies from pre-pregnancy

Several social, economic, cultural and behavioural factors contribute to low levels of health service utilisation in SSA. These include lack of transport and inaccessible health facilities[6]; high direct and indirect costs of care seeking such as fees, costs of food for mothers and accompanying children, new clothes appropriate to be seen in at ANC visits; the opportunity cost of time away from farming or other income generating activities[7 8]; and information asymmetry.[7]

Demand-side financing mechanisms such as cash transfers, in-kind transfers or voucher programmes have the potential to tackle financial and motivational barriers to care seeking and service utilisation.[9–12] Demand-side financing mechanisms have improved completion of tuberculosis treatment regimens[13 14]; reduced HIV risk, particularly among young women[15]; improved HIV testing, care and prevention,[16] and reduced the rates of illness among young children in low/middle-income countries.[17] In the context of reproductive, maternal, neonatal and child health, evaluated demand-side financing programmes have been effective in improving ANC attendance, and facility-based delivery.[18–22] None have explored targeted adherence to the continuum of care for ANC through to PNC and no economic evaluation of such programmes is available.

### The Afya trial

The Afya trial aims to test the effectiveness of a demand-side financing intervention to retain women in the continuum of care, from their first ANC visit until their children reach 1 year of age, in rural Kenya.[23] The intervention is a conditional cash transfer (CCT) payment for each facility appointment attended for ANC, facility-based delivery, PNC and childhood immunisation; and referrals related to any of these visits.

The intervention is being evaluated through a cluster randomised controlled trial in which 24 clusters are randomised to receive the intervention and 24 clusters are randomised to the control arm. The unit of randomisation is the health facility. The trial outcomes are the proportion of eligible ANC visits made by pregnant women, the proportion of women delivering at a health facility; the proportion of eligible health appointments attended for PNC; the proportion of expected immunisation appointments attended by children; and the proportion of health referrals for ANC, PNC and child immunisation attended. The trial is also evaluating and monitoring all aspects of the intervention process and implementation, including equity impact and the cost-effectiveness of CCT payments as a strategy to retain women in the continuum of care during pregnancy, birth and the postnatal period.

The Afya trial is described in detail in the trial protocol paper.[23] The purpose of this paper is to describe the protocol for the economic evaluation of the trial, comprising the cost-effectiveness and equity impact analyses.

### Economic evaluation of CCT payments for maternal and child health: what do we know?

Recent systematic reviews have found that CCTs have increased health service utilisation.[21 22 24] Glassman et al[21] reviewed the impact of CCTs on maternal and newborn health service use in eight South Asian and Latin American countries.[21] Their review found that these CCTs are associated with increased antenatal visits, births with skilled attendance and health facility delivery. A review by Chung[22] focused on evidence from seven sub-Saharan African countries and found mixed evidence on the utilisation of ANC and skilled attendance at delivery.[22] None of the studies included in either review present cost and/or cost-effectiveness of CCT programmes to improve maternal and child health. At the time of writing this paper, no stand-alone evaluations of these CCT programmes were found.

Hunter et al[24] synthesise evidence from seven published systematic reviews on the impact of conditional and unconditional cash transfers and vouchers on maternity service utilisation.[24] They find that cash transfers and voucher programmes can lead to an increase in the use of ANC, use of a skilled birth attendant and an increase in PNC. This review also identified three peer-reviewed studies of costs and cost-effectiveness of voucher programmes in South Asia and SSA but not of cash transfer programmes. One study examined the cost-effectiveness of a maternal voucher scheme in Bangladesh from the provider perspective and estimated the incremental cost per birth with a skilled attendant at US$69.85.[25] The second study conducted a cost-effectiveness analysis of the Makerere University Voucher Scheme in Uganda from the provider perspective. The intervention provided vouchers to pregnant women for transport and payment to service providers for ANC, delivery and PNC, and found the cost per birth was US$23.9 and the cost per PNC check-up was US$7.90.[26] The third study conducted a cost-effectiveness analysis of the same programme from the provider and societal perspectives and estimated the cost per disability-adjusted life year averted was US$302 and US$338, respectively.[27] These findings are unlikely to have direct relevance to the Afya trial, which focuses on CCTs and not vouchers. Voucher programmes encourage programme recipients to purchase and consume particular goods or services items, which is not the case with CCTs, where the cash received could expand the beneficiaries' budget set, thereby providing higher levels of utility.[28]

Thus, there is a scarcity of evidence on the cost and cost-effectiveness of CCT programmes. To our knowledge, this is the first study that assesses the cost-effectiveness of CCTs to improve maternal and child health (MNCH) in SSA and, as a result, this study will generate crucial evidence to inform the assessment of scale-up feasibility of CCT payments.

### Aim and objectives

The aim of the Afya economic evaluation is to estimate the cost and cost-effectiveness of a CCT programme to

retain women in the continuum of care during pregnancy, birth and the postnatal period in rural Kenya. The specific objectives of the economic evaluation are to:

1. Estimate the total direct and indirect costs of setting up and implementing the intervention, from the provider perspective.
2. Model the incremental cost-effectiveness of the intervention as compared with standard practice.
3. Model the expected cost of the intervention at scale in Kenya.
4. Analyse the equity impact of the intervention.

## METHODS AND ANALYSIS
### Study design
A cost-effectiveness analysis of the Afya trial will estimate the total and incremental costs and cost-effectiveness of the intervention prospectively from the provider perspective, measuring provider (programme and health service) costs.

### Study setting
Afya is being implemented in Siaya County, western Kenya. In 2016, Siaya County had an estimated population of 984 069 and a Human Development Index score of 0.46, significantly below the national average of 0.56.[29] The county has very poor indicators of maternal and child health.[30] Estimates from a 2011 survey show that the infant mortality rate was 111 per 1000 live births, far higher than the national rate of 49 per 1000. Similarly, the maternal mortality rate in Siaya is 695 per 100 000 live births, again much higher than the national maternity mortality rate of 488 per 100 000.[30] Data from a survey conducted in 2012 indicate that only 52% of mothers completed four WHO recommended ANC visits, and 49% delivered at a health facility, of which only 40% reported receiving any PNC services 48 hours after delivery. Only 18% of women reported all services along the continuum of care (ANC attendance, health facility delivery, PNC and newborn assessment) indicating low levels of service utilisation.[31]

In Kenya, health services are organised in six levels of care (table 1). In 2015 in Siaya, there were 174 health facilities of which 123 were public health facilities, 7 non-governmental, 16 faith-based and 28 private.[32] Among these, there is one level 5 facility and each subcounty has one level 4 facility.[33] The rest are mainly level 3 and 2 health facilities staffed by nurses or clinical officers, and level 1 community facilities staffed by community health volunteers. Overall, there is low coverage of healthcare, with a doctor to population ratio of 1:62 000, and nurse to population ratio of 1:2500.[32]

### Intervention and comparator description
All women recruited into the study are given an ANC clinic book as is standard practice and are provided with an enrolment card (henceforth, Afya card) attached to the clinic book. The Afya card is linked to a card reader installed at all participating health facilities. The Afya card is the size of a credit card and stores holder data such as authentication information (study ID, study arm, clinic at which enrolled) and pregnancy-related information (pregnancy stage at enrolment, expected delivery date, parity). The Afya card also allows visits to be tracked by touching the card on the card reader, which informs the payment (or not) of the CCT.

In the intervention arm, a CCT payment is made to pregnant women for each facility appointment attended for ANC, facility-based delivery, PNC and childhood immunisation; and referrals related to any of these visits. For each verified health visit made on time, a woman receives a cash transfer of KSh 450 (~US$4.5). In the control arm, women are also provided a nominal gratuity of KSh 50 (~US$0.5) in the form of mobile phone airtime to ensure that women carry the Afya card to ANC visits, health facility deliveries and PNC visits. The gratuity is transferred through the same system used to issue the incentives.

| Level | Type of facility | Health services offered |
|---|---|---|
| | **Table 1** Types of health facilities and health services offered, Kenya | |
| 1 | Community care | Facilitation of community diagnosis. Management; referral to higher level facilities; encouraging appropriate healthy behaviours. |
| 2 | Dispensary | Basic curative services; case management, prevention and promotion services; basic ANC services. |
| 3 | Health centre | Curative and case management services for infectious and chronic illnesses; inpatient care. |
| 4 | Subcounty hospital | Secondary care; primary care including ANC. |
| 5 | County referral hospital | Specialised services. |
| 6 | National referral hospital | Specialised diagnostic, therapeutic and rehabilitative services. |

Source: Ministry of Health (2017).
ANC, antenatal care.

## Trial design and study population

The unit of randomisation for the trial is a level 2 or 3 health facility. Level 2 and 3 health facilities are comparable and only those offering the full profile of ANC services were considered for inclusion in the study. The criteria for eligibility were: (1) women attending their first ANC visit; (2) long-term resident of the catchment area served by the health facility, with long-term residence defined as living in the area for at least 6 months; and (3) women with access to a mobile phone that belongs either to themselves or to a member of their household or person whom they trust. The total sample of enrolled women is 5488 women.

## Measurement of health outcomes

The primary outcomes of the trial are the proportion of eligible ANC visits made after recruitment; participants delivering at a health facility; eligible health appointments attended for PNC; expected immunisation appointments attended by children. We will test for differences between intervention and control arms in the primary outcomes using logistic regression for binary outcomes, and ordinal logistic regression for ordinal outcomes, adjusting for clustering using random effect models. Further details on the list of secondary outcomes and power calculations can be found in the trial protocol.[23]

## Identification, measurement and valuation of resource use

Cost and cost-effectiveness analyses will be conducted from the provider perspective, which takes into account the costs incurred by the provider in the provision of the health programme or intervention.[34] An overview of cost data is presented in table 2.

Provider costs are incurred by the institutions implementing the Afya intervention, namely Safe Water and AIDS Project, Kenya; the Stockholm Environment Institute, Sweden; and University College London, UK. These costs (programme costs, henceforth) data will be sourced from the financial project accounts of these institutions. These programme costs include the costs associated with starting up and implementing the intervention, which include but are not limited to the cost of CCT payments in the intervention arm; gratuity payments in the control arm; setting up, implementing, and maintaining the Afya card payment system; community and health facility sensitisation.

A step-down costing methodology will be used whereby costs from programme accounts are entered into a customised tool created in MS Excel.[35] The programme cost data are entered annually into the tool, which is adapted each year to reflect the changing cost structure of the trial at different phases of activity. Financial costs will be converted to economic costs, that is, any donated goods or volunteer time that do not appear in the programme accounting data will be added to the cost sheets and assigned a current market value.[34 36] Key informant interviews with programme leads will assist in identifying donated or subsidised items and in allocating joint costs between programme components. The allocation of joint staff costs is informed by monthly staff time sheets. Summary Excel worksheets present the costs by programme component and a single summary worksheet also summarises the total cost data, allows effect data to be entered and calculates the cost-effectiveness results. Research costs will not be included in the cost-effectiveness analyses. However, start-up costs will be reported and differentiated from implementation costs.

Provider costs are also incurred by the Ministry of Health in Kenya who provide ANC, facility-based delivery, PNC, child immunisation and related referral visits (health service costs). In the intervention arm, the CCT payment is likely to increase the demand for the ANC, facility-based delivery, PNC and immunisation services.

**Table 2** Description of costs, data sources and sample sizes

| Description | Type of cost | Description | Data sources | Sample size |
|---|---|---|---|---|
| Programme costs | Direct | Cost of implementing the Afya intervention | ► Project accounts of the implementation institutions.<br>► Key informant interviews with the project staff. | N/A |
| Healthcare service costs | Direct | Cost of visits made for ANC, delivery, PNC and immunisation | ► Health facility records and accounts.<br>► Key informant interviews with the facility managers. | 48 facilities |
| | Indirect | Opportunity cost of the increase in the workload of the facility staff, as the CCT payment is likely to stimulate demand for these health services | ► Key informant interviews with a subsample of facility staff. | A purposive sample of health workers in both arms (n=20) |

ANC, antenatal care; CCT, conditional cash transfer; PNC, postnatal care.

We will estimate any change in the demand for these services resulting from the intervention, and the concomitant value of any additional care provided. The routine monitoring of service statistics at the health facilities will capture the number of visits and any differences in visits between intervention and control arms will be attributed to the intervention. Primary data on the average unit cost of care will be collected from 48 facilities in the study area. A simple cost-capture form will be developed for facility data collection. The data from the facilities will be collected though interview with the facility manager, complemented with health facility records and accounts. Data from the cost-capture form will be used to complement existing data from centre reports, patients' records and published national and state reports relating to ANC, facility-based delivery, PNC, child immunisation and related referral visits.

## Economic evaluation

The economic evaluation of the Afya intervention will involve both cost-effectiveness analysis and cost-consequence analysis.

The cost-effectiveness analysis will be conducted as a within-trial analysis using the trial results. Results will be presented in terms of incremental cost-effectiveness ratios (ICERs), calculated as the arithmetic mean difference in cost between the intervention and control arms, divided by the arithmetic mean difference in effect. ICERs will be calculated for statistically significant primary outcomes as well as selected secondary outcomes, along with estimates of total cost at scale. Sensitivity analyses will be conducted to assess the impact on the cost-effectiveness, of changes in parameters with the greatest uncertainty, or with the greatest impact on the total costs. Cost-effectiveness acceptability curves will be generated to further describe uncertainty around the cost estimates.[37]

The results will also be presented as a cost-consequence analysis. All relevant costs and outcomes of the interventions will be listed in a tabular format, without aggregating into ratios. This allows policymakers to compare the incremental costs with the incremental consequences of different interventions. All statistically significant primary and secondary trial outcomes will be reported. Cost-consequence analysis is recommended for complex public health interventions, such as Afya, that have multiple health and non-health effects, which may be difficult to measure in a common unit.[34 38]

Costs will be presented in current prices in Kenyan Shillings and International Dollars (INT$). All costs will be adjusted for inflation using the consumer price index for Kenya and will be converted to 2020 INT$ using the 2020 purchasing power parity conversion factor for Kenya. Costs and outcomes will be converted to present values using an annual discount rate of 3% in the base-case, and annual rates of 0% and 6% in sensitivity analyses.

The equity impact of the intervention will be analysed within the economic evaluation to investigate whether the gains from such an intervention are equitably shared among the target population, that is, to investigate the extent to which different socioeconomic groups benefit from the intervention. The premise that underlies this component of the economic evaluation is that the intervention should disproportionately benefit the poorest, who also tend to have the highest need for health services.[39] The primary and secondary trial outcomes will be decomposed according to the socioeconomic status of participants. A multidimensional poverty index (MDPI) will be used to measure households' socioeconomic status. The MDPI allows us a more nuanced understanding of socioeconomic status. It takes monetary and non-monetary dimensions of deprivation into account and enables differentiation between population groups who may all be relatively poor in monetary dimensions such as income or asset ownership.[40] Thus, the consideration of other non-monetary attributes such as housing, literacy and so on in addition to income or asset ownership allows us to to distinguish between households that are homogenously asset or cash poor in this study setting.[41] Data at the household level on indicators in three dimensions of deprivation—health, education and living standards—will be collected during the enrolment survey from the trial participants. If a household is deprived if in a third or more of indicators, they are identified as 'MDPI poor'. The extent of household poverty is measured by the percentage of deprivations experienced, which also provides indications of relative poverty in this study setting.[42]

The affordability of the intervention will be explored using an analysis of fiscal space for programme delivery, a generalised fiscal space assessment method[43 44] and probabilistic analyses to determine a set of cost-effectiveness thresholds.[37 45] These analyses will also enable the exploration of a multicriteria decision analysis framework for resource allocation to this and other similar interventions to improve maternal and child health.[46] A wide range of affordability measures have been selected such as the total cost as a proportion of national gross domestic product (GDP), proportion of public health expenditure and proportion of public health expenditure on maternal and child health, in part, due to the paucity of evidence on the cost-effectiveness of comparable interventions.

## DISCUSSION

While there is evidence on the effectiveness of CCTs to improve the utilisation of maternal and child health services, there is little on the effectiveness and cost-effectiveness of CCTs to improve maternal and child healthcare in SSA. To our knowledge, this is the first study to assess the cost-effectiveness of CCTs to improve maternal and child healthcare in SSA. The protocol, which will adhere to internationally recognised guidelines for conducting and reporting economic evaluation studies, will provide transparency of planned data collection and economic evaluation analyses and improve the

rigour of the conduct, enabling greater comparability between findings.

The findings from this study will inform decision-makers about the value for money of this intervention, compared with others, and the fiscal space required to scale up the intervention in Kenya at a regional or national level, as well as in other settings where the utilisation of maternal and child health services is low.

### Patient and public involvement
Patients and public are not involved in the process of this economic evaluation study.

### DISSEMINATION
The results of the economic evaluation will be disseminated to the academic and policy-making communities, as well as the wider public, in a peer-reviewed journal and presented at a relevant international conference.

**Contributors** NB, HH-B, TP, AC, FV, AE, AM, CO and JS contributed to the study design. NB, HH-B, TP, AC and JS contributed to the analyses. NB, HH-B, TP, AO, SD and FV contributed to data collection and acquisition. NB was responsible for the initial drafting of this manuscript, and all authors contributed to the review of this manuscript and provided comments. All authors read and approved the final manuscript.

**Funding** The study is funded by the Bill and Melinda Gates Foundation, Funding ID OPP1142564.

**Competing interests** None declared.

**Patient consent for publication** Not required.

**Provenance and peer review** Not commissioned; externally peer reviewed.

**ORCID iDs**
Neha Batura http://orcid.org/0000-0002-8175-8125
Jolene Skordis http://orcid.org/0000-0002-8633-0208
Hassan Haghparast-Bidgoli http://orcid.org/0000-0001-6365-2944

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
