## [Reviewer comments · BMJ Open]

ARTICLE DETAILS

TITLE (PROVISIONAL)	Cost Effectiveness of Conditional Cash Transfers to Retain Women in the Continuum of Care during Pregnancy, Birth and the Postnatal Period: Protocol for an Economic Evaluation of the Afya trial in Kenya
AUTHORS	Batura, Neha; Skordis-Worrall, Jolene; Palmer, Tom; Odiambo, Aloyce; Copas, Andrew; Vanhuysse, Fedra; Dickin, Sarah; Eleveld, Alie; Mwaki, Alex; Ochieng, Caroline; Haghparast-Bidgoli, Hassan

VERSION 1 – REVIEW

REVIEWER	Yu Gao Charles Darwin University, Australia
REVIEW RETURNED	27-Aug-2019

GENERAL COMMENTS	The study aims to conduct an economic evaluation of a conditional cash transfer intervention to improve utilisation of maternity services in Kenya. The protocol clearly defined the resources to be included in the evaluation, perspective of the evaluation and the outcomes to be measured. Both cost-effectiveness and cost-consequences analysis will be conducted which I think is appropriate for a complex intervention. The authors also are planning to generate a cost effectiveness curves to describe the cost estimates uncertainty which is needed to assess the changes of parameters on he impact of the intervention cost-effectiveness. The study is also going to analyse the equity impact of the intervention to evaluate if the gains from the intervention are equitably shared among the target population, which is an important component and will give a comprehensive picture about the impact of the intervention.
--

REVIEWER	Margaret McConnell Harvard T.H. Chan School of Public Health, USA
REVIEW RETURNED	29-Aug-2019

GENERAL COMMENTS	This paper presents a protocol for a cost effectiveness analysis of CCTs for utilization of care during pregnancy, birth and the postpartum period. The paper describes plans for collecting costing data and performing cost effectiveness analyses. The proposed study is quite ambitious and I enjoyed reading the manuscript. The analysis described in this paper will be tremendously valuable for informing the potential scale-up of CCT programs in similar settings. However, the information provided in
---

	the paper is quite general and in several places the paper would be more useful if it provided additional details. I describe my suggestions for expanding the paper's contribution here. First, it would be helpful to provide additional theoretical justification for examining the equity implications of an analysis focused on a relatively homogenous rural population. What can be broadly learned about how equitably the impacts of the CCT program are distributed across more and less poor households within this family? Would it also be valuable to consider how the program impacts are distributed across other characteristics (for example such as geographic access or road quality)? Finally, additional details on how the multidimensional poverty index would be specified would also be valuable. The proposed cost effectiveness analysis uses very standard methods but does not very carefully consider or enumerate elements of costs that would be specific to a CCT program. For example, is there some consideration of costs associated with ensuring that households are informed about the existence of a CCT and the structure of the program? What would be the costs associated with maintaining sufficient data quality and ensuring that households or providers do not have the incentive to falsify records in order to receive the CCT? What are the costs associated with distributing and managing the transfers? A more thorough treatment of the specific program would be warranted. Finally, it would be helpful to more fully detail how participant's poverty score will be calculated and discuss related literature on measurement.
--	---

VERSION 1 – AUTHOR RESPONSE

Reviewer(s)' Comments to Author:

Reviewer: 1

The study aims to conduct an economic evaluation of a conditional cash transfer intervention to improve utilisation of maternity services in Kenya.

The protocol clearly defined the resources to be included in the evaluation, perspective of the evaluation and the outcomes to be measured. Both cost-effectiveness and cost-consequences analysis will be conducted which I think is appropriate for a complex intervention. The authors also are planning to generate cost effectiveness curves to describe the cost estimates uncertainty which is needed to assess the changes of parameters on the impact of the intervention cost-effectiveness. The study is also going to analyse the equity impact of the intervention to evaluate if the gains from the intervention are equitably shared among the target population, which is an important component and will give a comprehensive picture about the impact of the intervention.

Authors: Thank you for your positive feedback on the protocol for the economic evaluation of the Afya trial.

Reviewer: 2

This paper presents a protocol for a cost effectiveness analysis of CCTs for utilization of care during pregnancy, birth and the postpartum period. The paper describes plans for collecting costing data and performing cost effectiveness analyses.

The proposed study is quite ambitious and I enjoyed reading the manuscript. The analysis described in this paper will be tremendously valuable for informing the potential scale-up of CCT programs in similar settings. However, the information provided in the paper is quite general and in several places the paper would be more useful if it provided additional details. I describe my suggestions for expanding the paper's contribution here.

Authors: Thank you for your constructive feedback, we have responded to each of your suggestions below.

First, it would be helpful to provide additional theoretical justification for examining the equity implications of an analysis focused on a relatively homogenous rural population. What can be broadly learned about how equitably the impacts of the CCT program are distributed across more and less poor households within this family? Would it also be valuable to consider how the program impacts are distributed across other characteristics (for example such as geographic access or road quality)? Finally, additional details on how the multidimensional poverty index would be specified would also be valuable.

Authors: You make an excellent point about the relative homogeneity of this rural population. We have included a brief theoretical and policy-relevant rationale for the investigation of the equity impact of the intervention (see pages 13, lines 3-9). We also provide a more detailed justification of the use of the multidimensional poverty index and how this is constructed (page 13, lines 13-32). The indicators included in the index would be associated with other factors such as access to facilities that impact distributional effects and thus, the index captures these indirectly. That said, as you rightly point out, the findings from these analyses may be limited in their scope as the majority of this population might be considered 'vulnerable' or 'impoverished'. This limitation will be considered in full in the later paper presenting those results.

The proposed cost effectiveness analysis uses very standard methods but does not very carefully consider or enumerate elements of costs that would be specific to a CCT program. For example, is there some consideration of costs associated with ensuring that households are informed about the existence of a CCT and the structure of the program? What would be the costs associated with maintaining sufficient data quality and ensuring that households or providers do not have the incentive to falsify records in order to receive the CCT? What are the costs associated with distributing and managing the transfers? A more thorough treatment of the specific program would be warranted.

Authors: Thank you for this comment. Some of the CCT specific costs have now been listed on page 11 (lines 5-12).

Finally, it would be helpful to more fully detail how participant's poverty score will be calculated and discuss related literature on measurement.

Authors: We now provide more detail on the construction of the MDPI on page 13 (lines 13-32). A brief discussion on monetary and non-monetary dimensions of poverty measurement is now also included on page 13.

VERSION 2 – REVIEW

REVIEWER	Margaret McConnell Harvard T.H. Chan School of Public Health, USA
REVIEW RETURNED	15-Oct-2019
GENERAL COMMENTS	Thank you for the opportunity to review this revised manuscript. The revision addresses my suggestions for providing further clarity on design.